# Precise control of SCRaMbLE in synthetic haploid and diploid yeast

Bin Jia [1,2], Yi Wu[1,2], Bing-Zhi Li[1,2], Leslie A. Mitchell[3], Hong Liu[1,2], Shuo Pan[1,2], Juan Wang[1,2], Hao-Ran Zhang[1,2], Nan Jia[1,2], Bo Li[1,2], Michael Shen [3], Ze-Xiong Xie[1,2], Duo Liu[1,2], Ying-Xiu Cao[1,2], Xia Li [1,2], Xiao Zhou[1,2], Hao Qi[1,2], Jef D. Boeke[3] & Ying-Jin Yuan [1,2]

Compatibility between host cells and heterologous pathways is a challenge for constructing organisms with high productivity or gain of function. Designer yeast cells incorporating the Synthetic Chromosome Rearrangement and Modification by LoxP-mediated Evolution (SCRaMbLE) system provide a platform for generating genotype diversity. Here we construct a genetic AND gate to enable precise control of the SCRaMbLE method to generate synthetic haploid and diploid yeast with desired phenotypes. The yield of carotenoids is increased to 1.5-fold by SCRaMbLEing haploid strains and we determine that the deletion of *YEL013W* is responsible for the increase. Based on the SCRaMbLEing in diploid strains, we develop a strategy called Multiplex SCRaMbLE Iterative Cycling (MuSIC) to increase the production of carotenoids up to 38.8-fold through 5 iterative cycles of SCRaMbLE. This strategy is potentially a powerful tool for increasing the production of bio-based chemicals and for mining deep knowledge.

[1] Key Laboratory of Systems Bioengineering (Ministry of Education), School of Chemical Engineering and Technology, Tianjin University, Tianjin 300072, China. [2] SynBio Research Platform, Collaborative Innovation Center of Chemical Science and Engineering (Tianjin), Tianjin 300072, China. [3] Institute for Systems Genetics, New York University Langone Medical Center, 550 First Avenue, New York, NY 10016, USA. Correspondence and requests for materials should be addressed to Y.-J.Y. (email: yjyuan@tju.edu.cn)

The strategy of engineering heterologous pathways within host microorganisms has been used to improve the production of high-added-value biomolecules such as pharmaceuticals[1] and biofuels[2,3]. In general, incompatibility between the host cells and the heterologous pathway decreases productivity. Designer yeast cells incorporating the Synthetic Chromosome Rearrangement and Modification by LoxP-mediated Evolution (SCRaMbLE) system provide a platform for generating genotype diversity that can be followed by screening for advantageous organisms. The designer synthetic yeast chromosomes of Sc2.0 encode hundreds of loxPsym sites positioned downstream of non-essential genes and other major landmarks and thus are exquisitely sensitive to the expression of Cre recombinase. Previous studies have demonstrated that SCRaMbLEing the synthetic yeast chromosome generated genotype diversity including deletions, inversions, duplications, and other complex rearrangements[4–6]. Thus far, the synthetic yeast chromosomes synII, synIII, synV, synVI, synIXR, synX, and synXII have been fully synthesized and incorporated into *Saccharomyces cerevisiae* without major fitness defects[4,5,7–13], offering an opportunity to generate tremendously diverse host yeast strains that can be screened for the production of high-added-value biomolecules[14].

Precise control of the SCRaMbLE process is crucial for generating genotype diversity and organisms with specified advantages. Fusion of the estrogen-binding domain (EBD) to a variety of recombinases has been shown to confer ligand-dependent control of their activity[15]. Cre-EBD chimeras have been used to perform a wide range of ligand-dependent recombination reactions in mammalian cells and yeast[16,17]. Lindstrom and Gottschling[18] designed pSCW11-Cre-EBD for the separation of mother and daughter cells. The pSCW11 promoter is a daughter-cell-specific promoter that produces a pulse of recombinase activity exactly once in each cell's lifetime. In previous studies, SCRaMbLE was activated using a plasmid expressing pSCW11-Cre-EBD. However, growth defects in synthetic yeast containing the plasmid were observed in the absence of estradiol induction[4–6], suggesting that potential leaky expression of Cre recombinase decreased the stability of the synthetic chromosome. The selection of cells that have lost Cre plasmids would ensure increased stability of the synthetic chromosomes at the cost of more repetitions and potentially decreased diversity of the SCRaMbLEd population of cells. Moreover, it would be difficult to perform iterative cycles of SCRaMbLE with plasmid loss between each cycle. Therefore, it is necessary to construct genetic switches for the tight regulation of Cre recombinase to extend SCRaMbLE to larger-scale applications. Cheng et al.[19] characterized the galactose-driven Cre-EBD expression vector and showed that the EBD greatly reduced Cre-mediated recombination in the absence of ligand. Galactose-induced expression of the recombinase should result in rapid production of the target gene at levels comparable to those of the unaltered gene.

Here we construct a galactose-driven Cre-EBD as an AND gate in a genetic switch for the precise control of SCRaMbLEing synthetic haploid and diploid yeast. The AND gate combines transcriptional control and control of the cellular localization of Cre-EBD. We SCRaMbLE synV haploid yeast and synIII&V-bearing diploid yeast previously modified to carry a carotenoid-producing pathway. We develop Multiplex SCRaMbLE Iterative Cycling (MuSIC) to generate billions of combinatorial genomic rearrangement variants and increased their production continuously through iterative cycles of SCRaMbLE. We analyze SCRaMbLEd derivatives with improved carotenoid production by high-throughput sequencing and long-read sequencing, and we figure out the high-production phenotypes to the new genotypes.

## Results

**Design of AND gate switch for precise control of Cre.** Diverse genotypes and phenotypes can be generated through SCRaMbLE (Supplementary Fig. 1). To precisely control SCRaMbLE in *S. cerevisiae*, the AND gate pCRE4 (pGal1-Cre-EBD-tCYC1) was used to regulate Cre activity (Fig. 1a and Supplementary Fig. 2). The pGAL1 promoter is a galactose-inducible promoter and is tightly repressed by glucose[20]. Cre-EBD is a fusion of Cre recombinase and an EBD[18]. The expression of pCRE4 was designed as an AND gate (Fig. 1a), requiring the simultaneous addition of galactose and estradiol for full activity. Theoretically, Cre-EBD is unfolded and retained in the cytoplasm by binding to the Hsp90 chaperone in the absence of estradiol[21]. Leakiness of the activated Cre-EBD may delete essential genes in a synthetic chromosome and result in the loss of viability. To evaluate the leakiness of the two Cre switches, we compared the colony-formation activity of the synV yeast and synIII yeast containing the two Cre switches with that of the control (pRS413) in SC-His medium lacking both estradiol and galactose (Fig. 1a). Compared with the control, pCRE1 caused a subtle fitness defect in both the synV yeast and synIII yeast in the absence of estradiol, whereas pCRE4 had no observable effect on fitness. The subtle fitness defect in synV yeast containing pCRE1 might be due to small amounts of Cre-EBD escaping Hsp90 binding and entering the nucleus. pGAL1-Cre-EBD was an improved switch for the precise control of SCRaMbLE, because together the transcriptional control of the pGAL1 promoter and the localization control of Cre-EBD regulated the Cre recombinase very effectively (Fig. 1b). To assess the AND gate performance of the pCRE4 switch, the synV yeasts containing pRS413 and pCRE4 were cultured under four conditions as follows: (1) SC-His glucose medium without estradiol, (2) SC-His glucose medium with 1 μM estradiol, (3) SGal-His medium without estradiol, and (4) SGal-His medium with 1 μM estradiol (Fig. 1c and Supplementary Fig. 3). When cells were cultivated in SC-His glucose medium, there was no significant difference between pCRE4 and the control, regardless of whether estradiol was present, possibly because the Gal1 promoter is tightly repressed by the presence of glucose. However, a difference in fitness was observed when the cells were cultured in SC-His galactose medium. The doubling time (DT) of the synV yeast was increased by 8.3% compared with that of the control. This increase may be because the Gal1 promoter was highly induced with galactose and produced large amounts of Cre-EBD, small amounts of which escaped Hsp90 binding and entered the nucleus. When both galactose and estradiol were added to the media, the DT of the synV yeast was increased to 10 h, 1.5-fold higher than that of the control. Much more Cre-EBD protein was shown to localize in the nucleus and rearrange the genome upon combined induction with galactose and estradiol.

To study the expression strength of the two Cre switches, we constructed the pCRE5 (pSCW11-Cre-EBD-GFP-tCYC1) and pCRE6 (pGAL1-Cre-EBD-GFP-tCYC1) plasmids (Supplementary Fig. 4). Single-cell flow cytometric measurements of green fluorescent protein (GFP) (gfpmut3b) were also used to monitor the expression state of this control system (Fig. 1d). Low GFP expression values were observed for cell population containing pCRE5. Compared with pCRE5, cell population containing pCRE6 exhibited a dramatic increase in Cre-EBD-GFP expression, suggesting that the GAL1 promoter provided a higher expression level than the SCW11 promoter. Fig 1e displays GFP fluorescence in the cells. Low fluorescence levels were observed for pCRE5 in all three states. For pCRE6, no fluorescence was observed in the initial state and the switched-off state, whereas the strong emission from the switched-on state was clearly visible. In contrast to the SCW11 promotor, which is activated only once in the daughter cells, the Gal1 promoter results in constitutive

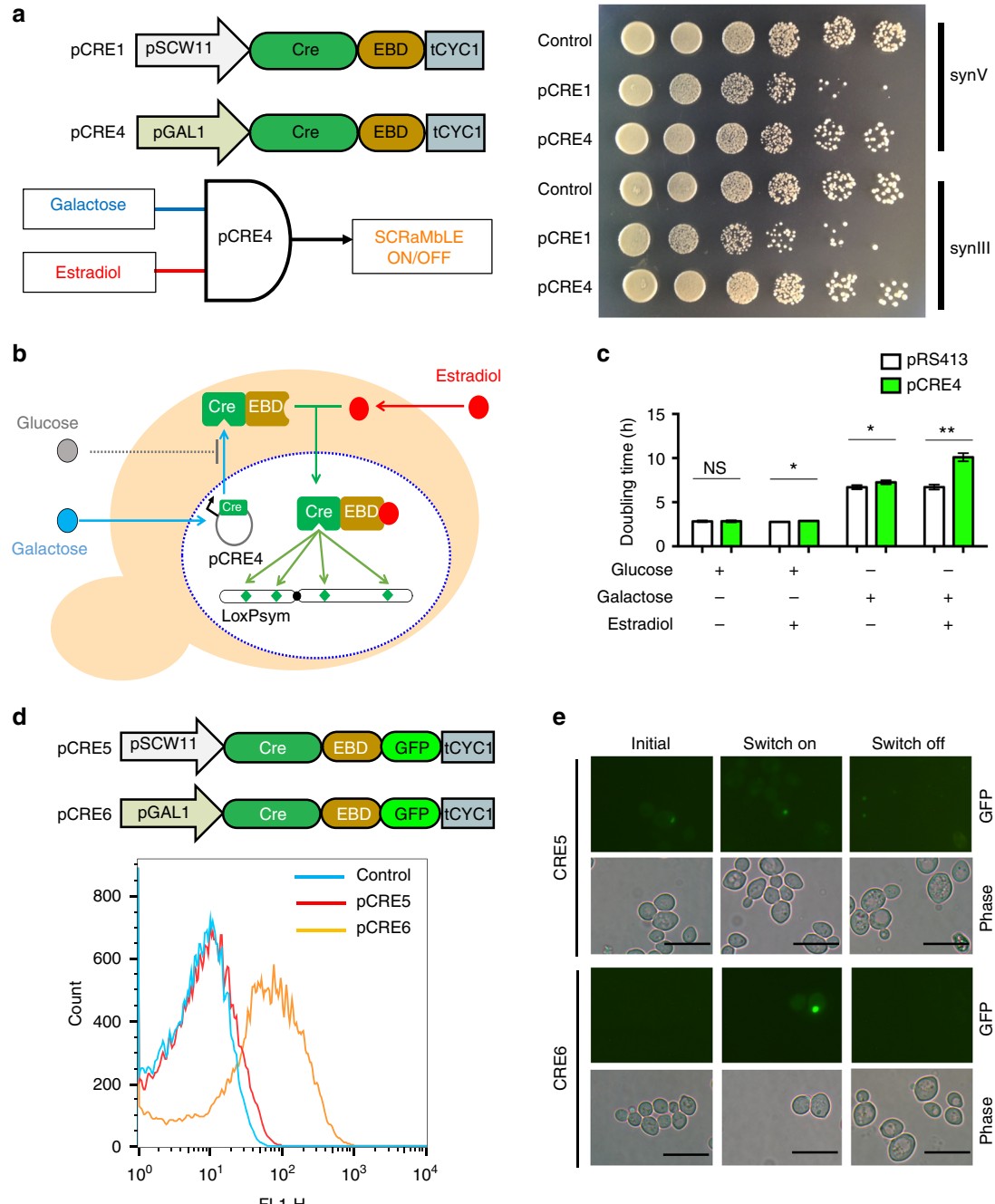

**Fig. 1** Design of the "AND gate" switch for precise control of SCRaMbLE. **a** Design of the AND gate for SCRaMbLE and fitness assays in synV yeast and synIII yeast, respectively. Activation of the "AND gate" pCRE4 required the input of galactose and estradiol simultaneously. Fitness assays of synV yeast and synIII yeast containing pCRE1, pCRE4, and control vector (pRS413) separately. Strain growth was assessed at 30 °C on SC-His medium. Shown here are 10-fold serial dilutions after 2 d growth. **b** Transcriptional regulation and cellular localization regulation of Cre recombinase for tight control of SCRaMbLE. pCRE4 was activated by induction with galactose and estradiol. **c** Comparison of the growth of synV strains containing pCRE4 and pRS413 under four conditions as follows: (1) SC-His glucose medium without estradiol, (2) SC-His glucose medium with 1 μM estradiol, (3) SGal-His galactose medium without estradiol, and (4) SGal-His galactose medium with 1 μM estradiol. (Student's *t*-test; NS, not significant; *$P \leq 0.05$, **$P \leq 0.001$). Error bars represent SD from three independent experiments. **d** Flow-cytometric assay expression level of Cre-EBD fusing GFPmut3b (488 nm excitation, 520 nm emission). Blue curves show fluorescence measurement of cells containing plasmids pRS413 incubated in SC-His glucose medium. Red curves show fluorescence measurement of cells containing plasmids pCRE5 incubated in SC-His glucose medium with 1 μM estradiol. Orange curves show fluorescence measurement of cells containing plasmids pCRE6 in SGal-His medium with 1 μM estradiol. **e** The "switch on" media for CRE5 and CRE6 were SC-His medium containing 1 μM estradiol and SGal-His medium containing 1 μM estradiol, respectively. The "initial state" and the "switch off" media were SC-His medium. The GFP fluorescence of the cells were imaged from this cell lawn using a fluorescence microscope (Olympus CX41). Scale bars, 10 μm

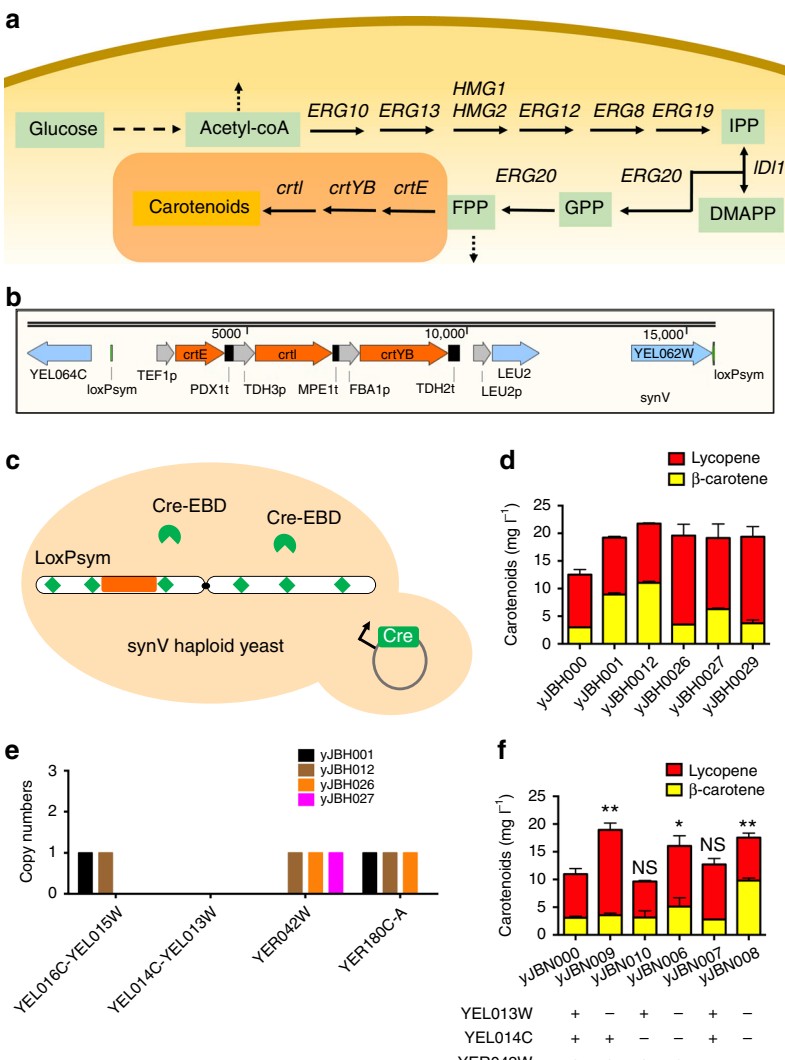

**Fig. 2** SCRaMbLEing synV haploid yeast with integrated carotenoid pathway. **a** MVA pathway in yeast and the carotenoid synthesis pathway. **b** The carotenoid pathway was integrated into the YEL063C/CAN1 site of the synV yeast. **c** The synV haploid yeast cells containing pCRE4 were induced to SCRaMbLE. Single colonies were cultured in 5 ml of SC-His overnight and then re-inoculated to an OD600 of 1.0 in 2% galactose SGal-His medium containing 1 μM estradiol for 8 h. **d** HPLC analysis of extracted carotenoids from cultures of five selected strains. **e** Deep-sequencing analysis of yJBH001, yJBH012, yJBH026, and yJBH027. Deletions of YEL014C-YEL013W were observed in synV of all the four haploid strains. **f** Reconfirmation of genome structural variation. yJBN000 is the control. yJBN006 is the YEL014C-YEL013W deletion of yJBH000. yJBN007 is the YER042W deletion of yJBH000. yJBN008 is the double deletion of YEL014C-YEL013W and YER042W in yJBH000. (Student's *t*-test; NS, not significant; *P < 0.05, **P < 0.001). In **d**, **f**, error bars represent SD from three independent experiments

expression in both mother and daughter cells in galactose cultures. Thus, the *Gal1* promoter potentially caused more cells to be SCRaMbLEd than the *SCW11* promotor by involving all the mother cells in SCRaMbLE.

**SCRaMbLEing the synV haploid yeast.** In yeast, acetyl-coA originates from glucose through the glycolytic pathway (Fig. 2a). The mevalonate (MVA) pathway converts acetyl-coA to farnesyl pyrophosphate (FPP; C15) through nine steps[22–26]. The heterologous carotenoid synthesis pathway consists of three genes, *crtE*, *crtI*, and *crtYB*[27,28]. Theoretically, SCRaMbLE generates diverse carotenoid phenotypes by genome rearrangement. As shown in Fig. 2b, we integrated the carotenoid pathway along with a Leu2 marker into the *YEL063C/CAN1* locus in synV to generate the yeast strain yJBH000. This integration of the carotenoid pathway into the *CAN1* locus had two advantages. First, *CAN1* is an endogenous genomic negative selectable marker and the use of

medium containing canavanine can select for the disruption of *CAN1* to verify the carotenoid pathway integration. As shown in Supplementary Fig. 5, the SC-Canavanine plate was used to select for carotenoid pathway integration. The colonies of synV haploid yeast with carotenoid pathway integration were orange, whereas no colonies appeared in the control group. Second, both *YEL063C* and *YEL062W* are non-essential genes that lie between two adjacent loxPsym sites. This unit can thus be freely deleted and duplicated without concern about linkage with other essential genes. This strategy allows the production of carotenoids, which leads to colored cells due to compound accumulation, for use in estimating the efficiency of SCRaMbLE.

Our AND gate switch was first applied to SCRaMbLE in the strain yJBH000 (Fig. 2c). A schematic diagram of SCRaMbLE is shown in Supplementary Fig. 6. To assay the relationships of colonies colors with the carotenoid production, five scrambled colonies exhibited different colors were inoculated on SC-His glucose medium and then dropped on an SC-His glucose agar

plate at 30 °C for 48 h. These five colonies were also performed fermentation experiments. The extraction and high-performance liquid chromatography (HPLC) analysis of carotenoid production are detailed in the Methods section. As shown in Supplementary Fig. 7, the five colonies with darkest color produced the highest carotenoids yield compared with other colonies of lighter color. It is indicated the color-base screen method can be used for high-throughput screening from the SCRaMbLE library. As shown in Fig. 2d, the carotenoid production of yJBH000 was 12.53 mg l$^{-1}$ and the carotenoid production of five SCRaMbLEd haploid strains (yJBH001, yJBH012, yJBH026, yJBH027, and yJBH029) was increased to 19.22, 21.76, 19.59, 19.15, and 19.38 mg l$^{-1}$, respectively. The SCRaMbLEd haploid strains increased the carotenoid yield 1.53- to 1.74-fold compared with the non-SCRaMbLEd parent strain.

To determine the genomic structural variation in haploids caused by SCRaMbLE, we deep sequenced yJBH001, yJBH012, yJBH026, and yJBH027, and aligned[29] the reads to chromosome synV. As shown in Fig. 2e and Supplementary Fig. 8, two deletions (YEL014C-YEL013W and YER042W) were observed in yJBH001. The YEL014C-YEL013W deletion was observed in yJBH012 and the YEL016C-YEL013W deletion was observed in yJBH026. Two deletions (YEL016C-YEL013W and YER180C-A) were observed in yJBH027. Long-read sequencing analysis of synV in yJBH001 showed the same two deletions without any inversions or translocations (Supplementary Fig. 9).

Interestingly, a common deletion (YEL013W-YEL014C) was observed in all four haploids. This finding strongly suggested that the YEL013W-YEL014C deletion is responsible for the improvement in carotenoid production. To analyze the effects of the two deletions on carotenoid yield, YEL013W, YEL014C, YEL014C-YEL013W, and YER042W were deleted from the control cells with a His3 marker. The resulting strains were named yJBN009, yJBN010, yJBN006, and yJBN007, respectively. The strain yJBN008 was made from yJBH000 through the deletion of both YEL014C-YEL013W and YER042W with a His3 marker and a KanMX marker, respectively. The strain yJBN000 was constructed from yJBH000 with a His marker integrated into the his3Δ1 locus. Compared with yJBN000, the carotenoid production levels of yJBN009, yJBN006, and yJBN008 were increased significantly, whereas those of yJBN010 and yJBN007 were not increased (Fig. 2f). The β-carotene percentage in yJBN008 was higher than that in yJBN009 and yJBN006. These results indicate that the YEL013W deletion increased carotenoid production, whereas the YEL014C deletion had no effect on carotenoid production. The deletion of YER042W increased the β-carotene percentage of carotenoid production in combination with the YEL013W deletion.

Transcriptome analyses of SCRaMbLE mutants were performed to assay the impact of SCRaMbLE (Supplementary Fig. 10). The yJBH012 had an obviously different pattern of global transcription as compared with the control yJBH000. The upregulation of 589 genes and downregulation of 236 genes were observed in this analysis (Log2FoldChange > 1-fold). Enrichment analysis of transcriptomics was used to identify differentially transcribed genes with known functions involved in KEGG pathways (Supplementary Table 1). Several genes involved in the glycolysis pathway were affected in the mutants. The upregulation of PGM2, FBA1, TDH1, TDH2, PGK1, ENO1, CDC19, and PDC1 increased the biosynthesis of acetyl-coA from glucose, which provided sufficient precursor for the MVA pathway[30–32]. In terpenoid backbone biosynthesis, the gene ERG10 were upregulated in the SCRaMbLEed mutants, suggesting increased flux of the MVA pathway[26,33]. As hydrophobic carotenoids accumulate in the lipid bodies of the cell, the upregulation of FAS1 and ACC1 increased the fatty acid biosynthesis, which increased cell storage

capacity of hydrophobic carotenoids[34,35]. It is indicated that transcription modification of those genes are responsible for the improvement of carotenoids production.

**SCRaMbLEing synIII&V diploid yeast.** As loxPsym sites are positioned downstream of non-essential genes, the deletion of large fragments containing essential genes in haploid yeast can result in loss of viability, potentially decreasing the diversity generated by SCRaMbLE. To overcome this challenge, diploids containing one copy of wild-type chrIII and chrV, as well as one copy of synIII and synV were SCRaMbLEd. This strategy allows the essential alleles in the wild-type chromosomes to remain intact. As shown in Fig. 3a, the synIII&V diploids were generated by mating synV yeast (MAT a) containing the carotenoid pathway with synIII yeast (MAT α). Multiple synthetic chromosomes will potentially increase the diversity of recombination events. As shown in Fig. 3b, five SCRaMbLEd strains exhibiting darker coloration were cultured and the carotenoids produced were extracted for quantification. The carotenoid production of the non-SCRaMbLEd strain yJBD000 was 0.96 mg l$^{-1}$. Five diploids exhibiting darker coloration (yJBD001 to yJBD005) produced 6.57, 7.52, 6.06, 8.38, and 6.66 mg l$^{-1}$ carotenoids, respectively. The SCRaMbLEd diploids increased the carotenoid production 6.29- to 7.81-fold compared with the control diploid. This result suggests that SCRaMbLEing diploid yeast can be used to increase the productivity of carotenoids.

Deep sequencing was performed to identify the SCRaMbLE events that occurred in the diploid strains and revealed three duplications (YEL072W-YEL071W, YEL070W-YEL060C, and YEL027W-YEL022W) and two deletions (YER033C-YER042W and YCR018C) in the diploid yeast yJBD001 (Fig. 3c and Supplementary Fig. 12). YER033C-YER042W is a 17.6 kb region containing two essential genes and seven non-essential genes. The absence of the synV YER033C-YEL042W segment and presence of the native wild-type chromosome V (V) YER033C-YEL042W segment was verified by the amplification of synthetic PCRTags (SYN) compared with wild-type PCRTags (WT) (Fig. 3d). Fitness assays comparing the yJBD001 strain with the synIII&V diploid yeast demonstrated that the deletion of the essential genes YER036C and YER038C did not result in a loss of viability (Fig. 3e). The YEL072W-YEL071W segment (7 kb) was increased to three copies, whereas the YEL070W-YEL060C segment (34 kb) and the YEL027W-YEL022W segment (16.7 kb) were increased to two copies. YEL070W-YEL060C is a 34 kb segment containing the carotenoid pathway; thus, the carotenoid pathway was duplicated by SCRaMbLE to produce two copies. The DNA copy numbers of the carotenoid pathway in the five improved diploids selected (yJBD001 to yJBD005) were assayed through quantitative PCR (qPCR) and the results showed this pathway duplication in four of the five diploids (Supplementary Fig. 13). This result suggests that duplication of the carotenoid pathway is very important for increasing the carotenoid production. To assay the effect of carotenoid pathway duplication, one carotenoid pathway was integrated into the CAN1 locus of the synIII haploid yeast (MAT α), which was then mated with yJBH000 (MAT a) to generate a new diploid (yJBN031) containing two copies of the carotenoid pathway. The carotenoid production of yJBN031 was increased 4.2-fold compared with that of yJBD000, but remained lower than that of yJBD001 (Fig. 3b). This result indicated that some other SCRaMbLE events coordinate with the carotenoid pathway duplication to improve carotenoid production.

To analyze the frequency of various SCRaMbLEd phenotypes, we divided the colony phenotypes into three visual categories: lighter-colored colonies, normal-colored colonies, and darker-colored colonies. The lighter-colored colonies likely displayed lower carotenoid productivity as a result of SCRaMbLE. In

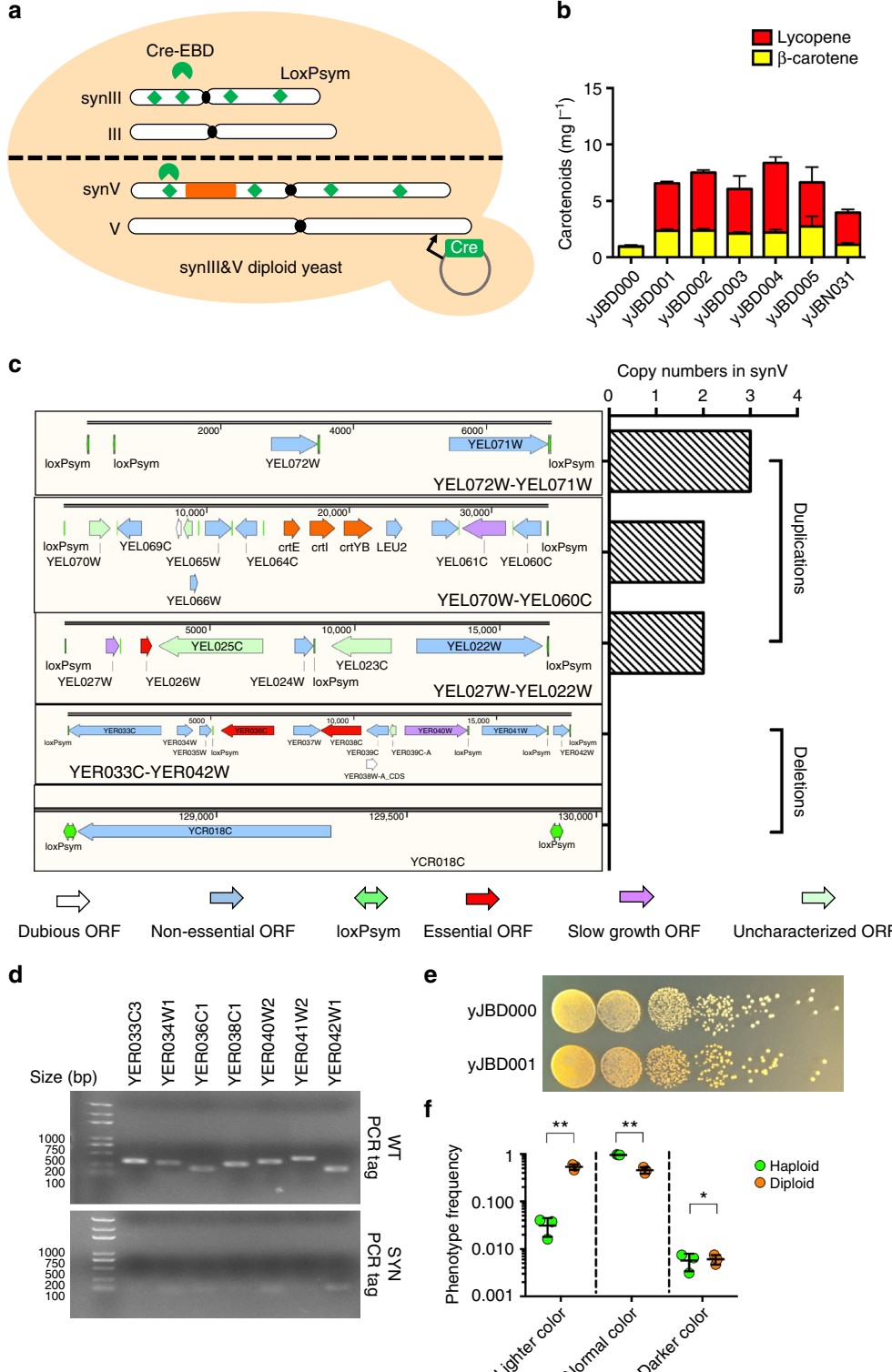

**Fig. 3** SCRaMbLEing synIII&V diploid yeast. **a** The synIII&V diploid yeast cells containing the pCRE4 were induced to SCRaMbLE. The synIII&V diploid yeast were generated by mating synV haploid yeast (*MAT a*) and synIII yeast (*MAT α*). **b** Quantification of extracted carotenoids from fermentation culture of strains. **c** Sequence analysis of yJBD001. Three duplications (YEL072W-YEL071W, YEL070W-YEL060C, and YEL027W-YEL022W) and two deletions (YER033C-YER042W and YCR018C) were observed in the diploid yeast yJBD001. YEL070W-YEL060C contains the carotenoid pathway. **d** YER033C-YEL042W PCRTag analysis. The absence of synV YER033C-YEL042W and the presence of native wtV YER033C-YEL042W was verified by the amplification of synthetic PCRTags (SYN) compared with that of wild-type PCRTags (WT). **e** Fitness assays of JSD3 strains compared with synIII&V diploid yeast. **f** Phenotype of diploid SCRaMbLE compared with that of haploid. The lighter color indicates SCRaMbLEd cells with a negative phenotype, including white colonies generated by deletion of the chromosome fragment containing the carotenoid pathway. The normal color indicates synthetic chromosomes with no rearrangement or no relevant rearrangement. The darker color indicates SCRaMbLEd cells with a positive phenotype. Bars are the mean of three biological replicates and error bars are the SD. (Student's *t*-test; *P < 0.05, **P < 0.001)

particular, white colonies were generated by deletion of the chromosome fragments containing enzymes in the carotenoid pathway (Supplementary Fig. 11). Normal-colored colonies potentially indicated either a lack of rearrangements or SCRaMbLE events with no correlation to carotenoid productivity. Darker-colored colonies indicated SCRaMbLE events resulting in higher carotenoid production. As shown in Fig. 3f, the phenotype frequencies of lighter-, normal-, and darker-colored post-SCRaMbLE haploid colonies were 3.14%, 96.6%, and 0.526%, respectively. The frequencies of lighter, normal and darker colored post-SCRaMbLE diploid colonies were 53.8%, 45.5%, and 0.61%, respectively. It is possible that the deletion of essential

and non-essential genes in haploids and diploids occur at equal frequencies. The deletion of essential genes in SCRaMbLEd chromosomes harbored in diploid strains could be tolerated without an obvious loss of viability. However, the deletion of essential genes in haploid strains is lethal and thus potentially decreased the frequencies of lighter- and darker-colored haploid colonies.

To analyze the genotype of yJBD001 more precisely, the tetrads of yJBD001 were dissected onto permissive agar plates[36] (Fig. 4a). Only one or two spores were observed to survive in each tetrad of yJBD001. The spores with synV in which YER033C-YEL042W was deleted most likely did not survive when dissected. Some of

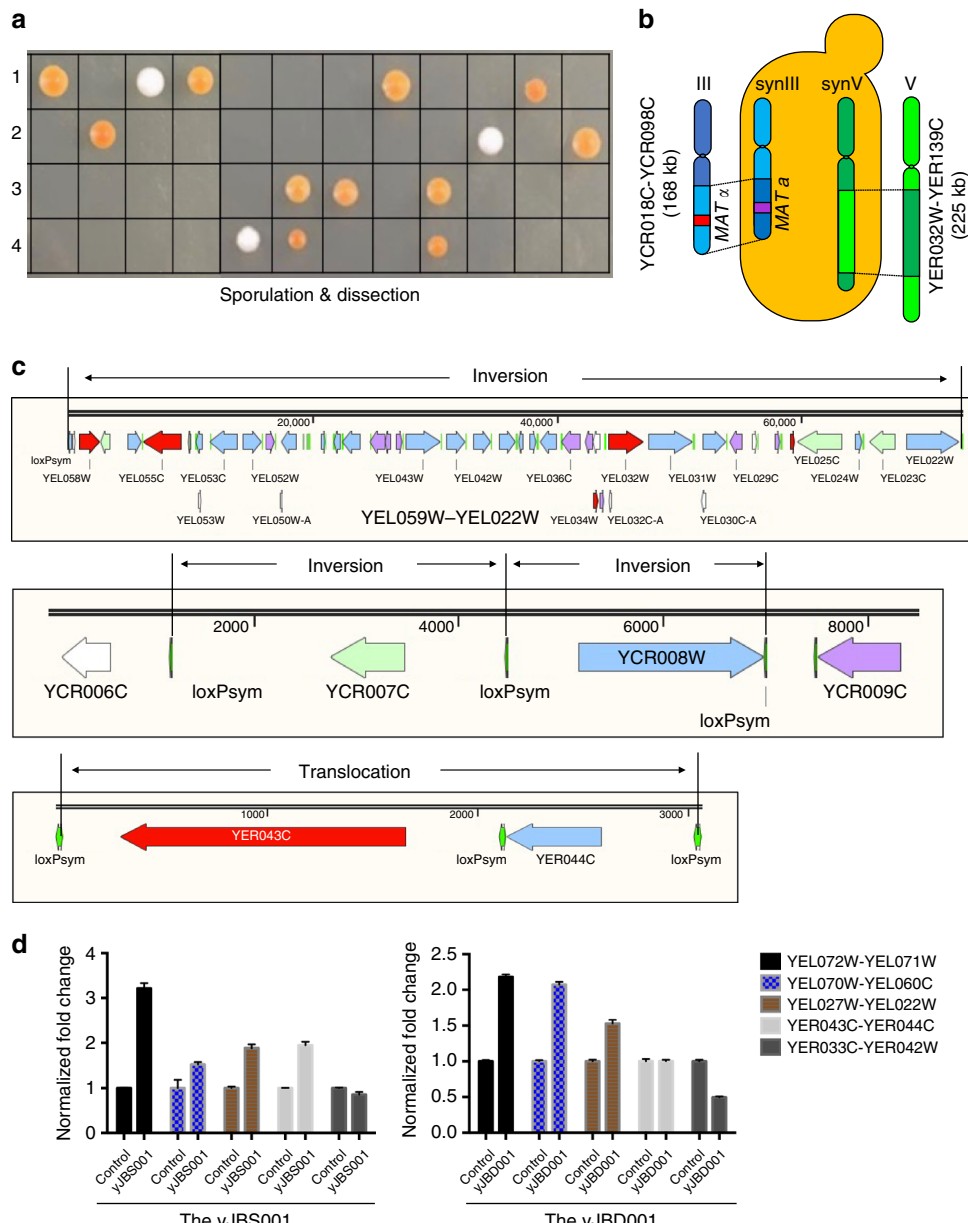

Fig. 4 Sporulation and deep sequencing analysis. a The diploid yJBD001 was sporulated and dissected on permissive agar plates, and the resulting haploid spore clones were analyzed to identify the mating type. b The crossing-over of two larger chromosome fragments was observed in the haploid spore yJBS001. The synIII YCR018C-YCR098C segment (168 kb) was exchanged with the wtIII YCR018C-YCR098C, and the YER032W-YER139C of synV (225 kb) was exchanged with the YER032W-YER139C of V. c Three inversions were observed in the haploid spore yJBS001, including one inversion in synV (YEL059W-YEL022W) and two inversions (YCR007C and YCR008W) in synIII. A translocation of YER043C-YER044C was observed in synV. d The copy numbers (duplication and deletion) of five locations in the diploid yeast yJBD001 and its spore yJBS001 were verified by qPCR. Error bars represent SD from three independent experiments

the spores were observed to become red or darker. The phenotypes of all spores were divided into three categories relative to yJBH000: white-colored spores, normal-colored spores, and darker-colored spores. We picked a darker-colored spore (yJBS001) and a normal-colored spore (yJBS003) to assay the production yield of carotenoids. As shown in Supplementary Fig. 14, the yields for yJBH000, yJBS001, and yJBS003 were 12.38, 33.46, and 15.38 mg l$^{-1}$, respectively. These results suggest that the main rearrangements responsible for increasing the carotenoid production yield of yJBD001 were segregated into yJBS001. To identify the rearrangement events, we used deep sequencing and long-read sequencing to analyze the genome of yJBS001. First, larger DNA segment crossing-over events were observed in both synV and synIII of yJBS001 (Fig. 4b and Supplementary Fig. 15). The segment containing *YCR018C-YCR098C* (168 kb) in synIII was exchanged with the segment containing *YCR018C-YCR098C* in the wild-type chromosome III. In addition, the segment containing *YER032W-YER139C* (225 kb) in synV was exchanged with the segment containing *YER032W-YER139C* in wild-type V, which was responsible for the survival of the spores containing the main synV. The chromosomal crossing-over events between homologous SCRaMbLEd synthetic chromosomes and wild-type chromosomes resulted in recombinant chromosomes, which further increased the genomic diversity. Second, three inversions were observed though long-read sequencing (Fig. 4c and Supplementary Fig. 16). *YEL059W-YEL022W* was a large segment (73 kb) of inversion in synV, whereas *YCR007C* and *YCR008W* underwent inversion independently even though they were adjacent in synIII. Lastly, four duplications of DNA segments in synV were observed in the haploid spore yJBS001 (Supplementary Fig. 15). The *YEL072W-YEL071W* segment, the YEL070W-YEL060C segment, and the *YEL027W-YEL022W* segment were increased to 3, 2, and 2 copies, respectively, which are the same as the numbers in yJBD001. The *YER043C-YER044W* segment (3 kb) was likewise increased to two copies. To assess these duplications in the spore (yJBS001) and the diploid (yJBD001), we used qPCR to analyze the DNA copy numbers of four genes (*YEL071W*, *CrtE*, *YEL022W*, and *YER043C*) that were representative of the duplicated segments (Fig. 4e). For yJBS001, the fold change in the copy numbers of the *YEL072W-YEL071W*, *YEL070W-YEL060C*, *YEL027W-YEL022W*, and *YER043C-YER044W* segments were nearly the same whether assessed with qPCR or whole genome sequencing, using yJBH000 as the control. However, for yJBD001, the fold change in the copy numbers of the *YEL072W-YEL071W*, *YEL070W-YEL060C*, *YEL027W-YEL022W*, and *YER043C-YER044W* segments were 2:1, 2:1, 1.5:1, and 1:1, respectively, using yJBD000 as the control. The carotenoid pathway was increased to two copies along with the synthetic segment *YEL070W-YEL060C*, and the total copy numbers of the *YEL072W-YEL071W* and *YEL027W-YEL022W* segments were 4 and 3, respectively. The crossing-over segment (*YER032W-YER139C*) contains the *YER043C-YER044C* and *YER033C-YER042W* segments, suggesting that the *YER043C-YER044C* segment was translocated to another locus outside the crossing-over segment (*YER032W-YER139C*).

To analyze the phenotype frequencies of spores, more tetrads were dissected on 60 agar plates (Supplementary Fig. 17). Darker-colored spores were screened from 46 (76.7%) of the total 60 plates, and the darker color phenotype ranged from 0 to 38.5% in a single plate. In summary, 86 red spores (11%), 256 white spores (33%), and 442 orange spores (56%) were observed from a total of 784 spores on the 60 plates, which was consistent with the law of linkage and crossing over. This result indicates that the sporulation and dissection of SCRaMbLEd diploid yeasts might be a productive approach to enhancing desired phenotypic traits.

**Multiplex SCRaMbLE Iterative Cycling**. To rapidly and continuously generate chromosome diversity across a large population of cells, we developed MuSIC, which is used to iteratively accumulate multiple rearrangements through multiple cycles of SCRaMbLE (Fig. 5a). Through step-by-step screening accumulation, cells with improved production yields are grown to mid-log phase in SC-His liquid medium for the subsequent cycle. High-production diploid yeasts can be dissected to screen for high-production spores, which are capable of mating with other synthetic yeast haploids for the subsequent cycle of SCRaMbLE.

Key concerns for this process included the stability of the SCRaMbLE switch when cultured in different mediums as well as its scalability in long-term culture for high-throughput screening. To assay the potential leakiness of the pGAL1-Cre-EBD in media with different carbon sources, yJBD000 strains containing pCRE4 were cultured for 24 h in SC-His medium containing 2% glucose, sucrose, galactose, potassium acetate, ethanol, or glycerol, separately (Supplementary Fig. 18). White colonies were observed on the galactose medium plate, indicating that pGAL1-Cre-EBD was not appropriate for work related to galactose utilization. To assess the stability of our AND gate for long-term culture, yJBH000 cells bearing pCRE4 were first incubated in SC-His glucose medium for 48 generations and subsequently re-incubated in SGal-His medium containing estradiol to induce SCRaMbLE (Supplementary Fig. 19). We demonstrated that yJBH000 cells grown in the "switch-off" state of the AND gate did not display an obvious growth defect during 6 days of subculturing, whereas subsequent induction of the "switch-on" state of the AND gate resulted in a significant growth defect in yJBH000. This result proved the suitability of pCRE4 for long-term culture.

Second, to facilitate the visual screening process for strain populations with high carotenoid production, we incubated the SCRaMbLEd yeast plates at 30 °C, 33 °C, 35 °C, and 37 °C (Supplementary Fig. 20). Significant fading of colony color was observed at 37 °C. Incubation at 33 °C and 35 °C resulted in a slight shift from darker orange to yellow. This experiment demonstrated that incubation at high temperature can reduce the background color of a SCRaMbLEd population and facilitate visual screening for yeast that produce a higher yield of carotenoids[37,38]. Shi et al.[37] demonstrated that low temperature was more appropriate for beta-carotene production in yeast, whereas high temperature decreased carotenoid synthesis in yeast. Moreover, high temperature affected the cellular metabolism network of yeast, which also decreased the carotenoid production.

As a proof of concept, yJBD001 (6.57 mg l$^{-1}$ carotenoids) was used for iterative cycles of SCRaMbLE. As shown in Fig. 5b, yJBD001, yJBD038, yJBD048, yJBD057, and yJBD069 were generated by SCRaMbLEing yJBD000, yJBD001, yJBD038, yJBD048, and yJBD057, respectively. The carotenoid production levels of yJBD038, yJBD048, yJBD057 and yJBD069 were increased to 12.37 mg l$^{-1}$ (12.8×), 29.98 mg l$^{-1}$ (31.1×), 35.83 mg l$^{-1}$ (37.2×), and 37.39 mg l$^{-1}$ (38.8×), respectively, representing a significant fold change from the production of yJBD000. The phenotype frequencies of darker-color colonies appearing after cycle 2, cycle 3, cycle 4, and cycle 5 were 0.67%, 0.69%, 0.48%, and 0.63%, respectively (Supplementary Table 2). A larger increase in productivity occurred during the first four cycles and only a slight productivity increase in the last cycle, perhaps caused by reaching the maximum ability of SCRaMbLE to improve desired phenotypes or by a growth defect resulting from high carotenoid accumulation (Supplementary Fig. 21).

The mating type of synX yeast (*MAT a*) had been shifted to *MAT α*[39]. The new diploid yJBD200 containing synX, synIII, and synV was generated by mating yJBS001 and synX yeast.

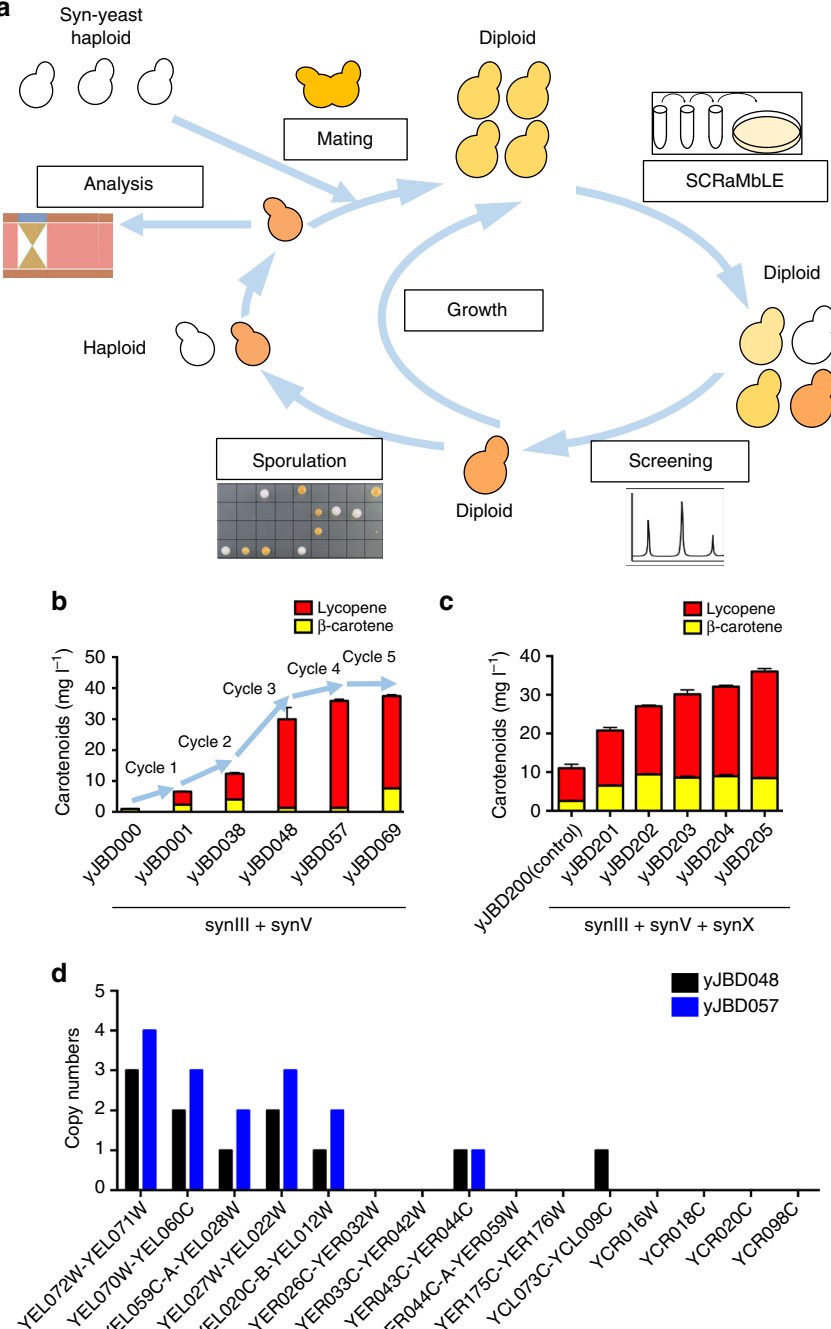

**Fig. 5** MuSIC strategy for rapid evolution. **a** MuSIC enables the generation of chromosome diversity across a large population of cells through the iterative SCRaMbLEing and screening of yeast. Based on step-by-step screening and accumulation analysis, cells with improved production are then recovered for the subsequent cycle. High-production diploids are dissected to screen for high-level production spores. The spores are capable of mating with other synthetic haploid yeast for subsequent cycles. **b** Five cycles of MuSIC to screen for high-production diploids. The yJBD001, yJBD038, yJBD048, yJBD057, and yJBD069 strains were generated by SCRaMbLEing the yJBD000, yJBD001, yJBD038, yJBD048, and yJBD057 strains, respectively. **c** Incorporation of multiple synthetic chromosomes for SCRaMbLE. The spore from dissection of yJBD001 was mated with synX yeast haploids for the subsequent cycle of SCRaMbLE. **d** Deep sequencing analysis of yJBD048 and yJBD057. Duplication of YEL014C-YEL013W was observed in synV of yJBD057. In **b**, **c**, error bars represent SD from three independent experiments

SCRaMbLEing yJBD200 can also generate high-carotenoid-production strains. As shown in Fig. 5c, compared with the carotenoid production level of yJBD200 (11.03 mg l$^{-1}$), those of yJBD201, yJBD202, yJBD203, yJBD204, and yJBD205 were increased to 20.80, 27.06, 30.11, 32.11, and 36.03 mg l$^{-1}$, respectively. This result suggests that incorporation of more synthetic chromosomes potentially increased the diversity of genome rearrangement and carotenoid-production yield.

In addition, we analyzed two high-carotenoid-production SCRaMbLEd diploids (yJBD048 and yJBD057). As shown in Fig. 5d, deep sequencing of the yJBD048 strain revealed two deletions in synV (*YER026C-TER059W* and *YER175C-YER176W*) and four deletions in synIII (*YCR016W*, *YCR018C*, *YCR020C*, and *YCR098C*) compared with yJBD001 (Supplementary Fig. 22). Deep sequencing of the yJBD057 strain revealed that yJBD057 had a much larger duplication (130 kb) from *YEL072W* to

YEL012W and a larger deletion in synIII (YCL073C-YCL009C) than those in yJBD048 (Supplementary Fig. 23). To assay the chromosome stability, yJBD057 and yJBD000 strains were serially subcultured for 5 days and plated on Yeast Peptone Dextrose (YPD) agar (Supplementary Fig. 24). No white colonies were observed on yJBD000 plates, whereas white-colored colonies (0.68%) were observed on yJBD057 plates. PCRTag analysis was used to assay stability of the synV chromosome. As shown in Supplementary Fig. 24b, no additional losses were observed in yJBD000 and yJBD057 colonies, respectively. Deletions of YEL072W-YEL060C were observed in white colonies arising from yJBD057. As high-copy duplications of the YEL072W-YEL060C region were observed in yJBD057 (Supplementary Fig. 23), deletions of YEL072W-YEL060C may be caused by self-homologous recombination. It is concluded that the synthetic chromosomes were stable after 60 generations, and the inclusion of lox sites will not ruin strain stability. Two possible reasons exist. First, previous works have demonstrated the stability of synthetic chromosomes and have yet to detect spontaneous deletions mediated by loxPsym sites[4,5,7,8,11–13]. Second, because high-level carotenoids production in S. cerevisiae leads to membrane stress and is toxic for the producing cell[28,40]. Evolutionary pressure likely caused inactivation of the heterologous pathways in a few cells by spontaneous recombination or mutagenesis[41–43]. From these experiments, we found the deletion rate of the transplanted carotenoid pathway to be 0.68% after 60 generations.

## Discussion

Our AND gate enabling precise control of SCRaMbLE allows double regulation on both a transcriptional and a cellular localization level. High stability and reliability enable the iterative SCRaMbLEing of synthetic chromosomes without removing the Cre plasmid. It is noted that pCRE4 was observed to exhibit leakiness when strains were cultured in galactose medium (Supplementary Fig. 18), pGAL1-Cre-EBD is not suitable for galactose utilization studies. There are many novel inducible promoters in yeast[44,45] that can be used to derive Cre-EBD expression. The AND gate strategy[46] should be a good approach for decreasing leakiness in many different applications.

Our study indicates that SCRaMbLE can be used for the genome-wide mining of new knowledge. It is well known that the overexpression of rate-limiting enzymes in a pathway and repression of the competing pathway can effectively improve the supply of FPP[26,47,48]. However, our method can also involve global gene rearrangements, especially the deletion or duplication of genes outside the pathway. Our study proved that the YEL013W deletion can increase the improvement in carotenoid. YEL013W is a phosphorylated and palmitoylated vacuolar membrane protein; the vacuoles of YEL013W deletion cells are multilobular or even fragmented into small vesicles, and the cytoplasm-to-vacuole transport pathway is strongly impaired[49]. Deletion of YEL013W is indicated to affect the regulation of autophagy in yeast, which caused global transcription modification (Supplementary Fig. 10). Upregulation of endogenous genes and pathways increased the carotenoid production (Supplementary Table 1). The YEL014C is a putative gene of unknown function, and the deletion of YEL014C has no significant effect on carotenoid production. YER042W is a methionine-S-sulfoxide reductase that participates in the response to oxidative stress and catalyzes the reduction of dimethyl sulfoxide to dimethyl sulfide in an NADPH-dependent manner[50]. The deletion of YER042W potentially affects the cellular cofactor balance and thus alters the carotene/lycopene ratio in high-production strains. It is noted that none of those genes were reported to affect the MVA

pathway directly. Therefore, it provides a method to rapidly mine new knowledge. Moreover, in this study, 6.07% deletion (32,520 bp) of synV (536,024 bp) was observed in one cycle of SCRaMbLEing haploid yeast (Supplementary Fig. 11). Genome minimization is an attractive approach for investigating the core functions of life and for exploring whole-genome design[51,52]. Our SCRaMbLE platform might be used to simplify or even minimize the entire yeast genome rapidly.

In this manuscript, deep-sequencing analysis of the boundaries of the recombination events indicate that all rearrangements were precise recombinations at designed loxPsym sites (Supplementary Fig. 25). All nine genes of the MVA pathway were present in nonsynthetic chromosomes (Supplementary Table 3). Deep sequencing of the SCRaMbLEd yeast revealed that no single-nucleotide polymorphisms and no duplications were observed in any of the nine genes (Supplementary Fig. 26), and no ectopic recombinations were observed in the remaining nonsynthetic genome. These results are consistent with those of Shen et al.[6]. However, fewer recombination events occurred than those in Shen's experiments[6]. Three possible reasons exist. First, the synIXR used by Shen is a circle bacterial artificial chromosome and replicates by rolling circle amplification, which might have generated more recombination events than the replication of the linear synthetic chromosome (synV) during SCRaMbLE. Second, synV contains 41 essential genes, whereas synIXR contains only 4 essential genes. Essential genes are more easily deleted from synV than from synIXR during SCRaMbLE, which decreased the high number of recombination events in synV. Finally, the synIXR SCRaMbLE colonies were selected for auxotrophies arising from loss of function of LYS1 or MET28 encoded in the synthetic region, whereas the SCRaMbLE strains in the manuscript were selected for high carotenoid production.

To increase the types and amounts of SCRaMbLE, this study developed diploid SCRaMbLE and proved it to be more effective than the use of haploid yeast to preserve SCRaMbLE diversity. SCRaMbLEing haploid cells results in a severe loss of viability[53], which theoretically decreases the diversity of the resulting population. However, as heterozygous diploids have backup chromosomes to maintain the core functions of life, genotype diversity can be stably inherited. Therefore, large numbers of DNA rearrangements were observed in diploid SCRaMbLE, including the deletion of essential genes and duplications of larger numbers of essential and non-essential genes on the synthetic chromosomes. The sporulation and dissection of diploids might further promote genome rearrangement by crossing over. The crossing over (YER032W-YER140W) was not flanked by loxPsym sites, indicating that the crossing over is independent of the loxPsym sites. The combination of SCRaMbLE and crossing over can be used to generate more complicated genome rearrangements.

In this work, the ctrE, crtI, and CrtYB genes were amplified from the complementary DNA of Xanthophyllomyces dendrorhous. Verwaal et al.[28] constructed an S. cerevisiae strain (YB/I/E + tHMG1 + I) capable of producing 5.9 mg g⁻¹ dry cell weight (DCW) of β-carotene. The GAL regulatory system was used to drive the expression of YB/I/E + tHMG1 and achieved the production of 11 mg g⁻¹ DCW of total carotenoids[54]. The carotenoid yield of the synIII + synV heterozygous diploid was increased from 0.22 mg g⁻¹ DCW to 12.71 mg g⁻¹ DCW after five cycles of MuSIC. The carotenoid pathway was increased to three copies in adjacent regions of synV in the high-production strains. The evolutionary genotypes indicate that increasing the heterologous carotenoid pathway expression is still important for improving carotenoid production. Many methods are available to increase the stability of heterologous carotenoid pathway expression. For example, inducible promoters with strong transcription ability

can be used to increase the expression of carotenoid pathway[54–56]. Tuning antibiotic concentrations to increase the plasmid copy numbers can also result in high pathway expression[33].

In addition, three synthetic chromosomes (synX + synIII + synV) can generate higher carotenoid production than two synthetic chromosomes (synIII + synV) in one cycle of SCRaMbLE. This result suggests that involving more synthetic chromosomes in SCRaMbLE will increase the rearrangement of potential target genes and thus optimize the compatibility between host cells and heterologous pathways. The integration of other synthetic chromosomes (synII, synVI, and synXII) or even all 16 synthetic chromosomes is anticipated to yield a powerful platform for generating thousands of combinations of DNA segment rearrangements through MuSIC. Deep sequencing and further study of the relationship between genotype and phenotype will help us to better understand the metabolic network and engineered yeast chassis for larger-scale industry applications. Although the example we used was the carotenoid pathway, many endogenous or exogenous pathways are associated with cognate biosensors[57–60] and could be assessed using our MuSIC strategy by changing the targeting and screening design.

In summary, our AND gate enabling the precise control of SCRaMbLE provides new insights and solutions for increasing the productivity of cell factories and extending the limits of our biological knowledge.

## Methods

**Strains and media**. Yeast strains are described in Supplementary Table 4. Plasmid cloning work and circuit construct characterization were all performed in *Escherichia coli* DH10B strains, which were cultured in LB (Luria–Bertani broth) media (10 g l$^{-1}$ peptone, 5 g l$^{-1}$ NaCl, 5 g l$^{-1}$ yeast extract). *S. cerevisiae* strain BY4741 (*MATa his3Δ1 leu2Δ0 met15Δ0 ura3Δ0*) was used as the initial host for most DNA assembly and transformation in this study. Yeast strains were cultured in YPD medium (10 g l$^{-1}$ yeast extract, 20 g l$^{-1}$ peptone, and 20 g l$^{-1}$ glucose), and SC-His (synthetic complete medium lacking histidine with 20 g l$^{-1}$ glucose) and (synthetic media lacking histidine with 20 g l$^{-1}$ galactose); β-Estradiol and ʟ-canavanine were purchased from Sigma-Aldrich. SC-Canavanine plate (synthetic complete medium lacking arginine with 20 g l$^{-1}$ glucose and 60 μg ml$^{-1}$ ʟ-canavanine) were used to select for pathway integration CAN1 locus. Presporulation medium (50 g l$^{-1}$ glucose, 30 g l$^{-1}$ Difco nutrient broth and 10 g l$^{-1}$ Difco yeast extract), sporulation media (10 g l$^{-1}$ potassium acetate, 0.05 g l$^{-1}$ zinc acetate dehydrate, add uracil, histidine, leucine at 1 × concentrations), and 1 × stock solution of zymolyase (50 μg ml$^{-1}$ in 1 M sorbitol) were prepared for sporulation and tetrad dissection.

**Plasmid circuits construction**. Plasmids are described in Supplementary Table 5. All plasmids were constructed using standard molecular cloning techniques and Gibson assembly. All plasmid maps are shown in the Supporting Information. Restriction endonucleases, T4 DNA Ligase and Phusion PCR kits were used from New England BioLabs. PCRs were carried out with a ABI Thermal Cycler. Primers were synthesized by Genewiz. All plasmids were transformed into *E. coli* strain DH10B with standard protocols and isolated with TIANprep Mini Plasmid Kits. Plasmid constructs were verfied by restriction digests and sequencing by Genewiz. The CRE, CRE-EBD, and the CYC1t were PCR amplified from the pCRE1. The promoter ZEO1p and GAL1p were PCR amplified from genome of BY4741. ZEO1p-CRE-EBD-CYC1t and GAL1p-CRE-CYC1t and GAL1p-CRE-EBD-CYC1t were assembled into pRS413 linearized by SalI and BamHI through Gibson method[61]. The pRS416-carotenoid was constructed using yeast assemble method[62]. Specifically, The TEF1p, tTDH3p, tFBA1p, and the TDH2t were PCR amplified from genome of BY4741. The *crtE*, *crtI*, and *crtYB* were PCR amplified from the Registry of Standard Biological Parts (http://partsregistry.org). pRS416 was linearized by EcoRI digestion. After gel purification, all the eight DNA fragments were co-transformed into BY4741. Plasmids extracted from orange color colonies were transformed into DH10B.

**Yeast transformation and assembly**. The protocol for yeast transformation is the LiAc/SS carrier method[63]. Yeast colonies were inoculated into 5 ml of YPD and grown overnight at 30 °C. Cultures were pelleted and washed twice with 0.1 M of LiAc solution. Four hundred and eighty microliters of 50% polyethylene glycol (PEG) with molecular weight 3350, 40 μl salmon sperm DNA (100 mg ml$^{-1}$), 72 μl of LiAc solution (1 M) and plasmid (1 mg ml$^{-1}$) were mixed. Then 100 μl cells was resuspended in LiAc/SS carrier DNA/PEG mixture. Samples were first incubated at 30 °C for 30 min and then heat-shocked for 15 min at 42 °C. After discarding the PEG mixture, pellets were resuspended in 400 μl dH₂O and plated on selective medium agar plates. For yeast assembly, 200 ng of each gel purified fragments and the linearized vector were mixed together. Following transformation, yeast colonies were selected on synthetic medium lacking histidine (SC-His) agar plates for 3 days at 30 °C.

**Doubling time calculation**. The strains were inoculated in SC-His medium overnight at 30 °C and 220 r.p.m. Cultures were back diluted to OD600 = 0.1 in appropriate media. Then, 200 μl of cell culture was transferred to a 96-well fluorescence plate and incubated at 30 °C and the optical density was read at 600 nm every 2 h. The DT was calculated using the formula:

$$\mathrm{DT} = \mathrm{DurationTime} * \log 2 * [\log(\mathrm{FinalConcentration}) - \log(\mathrm{InitialConcentration})]^{-1}$$

where "log" is the base logarithm. The Initial Concentration point and the Final-Concentration point were chosen from the exponential growth phase. Online DT computation was performed at http://www.doubling-time.com/compute.php.

**Fluorescence assay**. The cellular Cre-EBD-GFP expression level was quantitatively assayed. pCRE5 and pCRE6 were constructed by fusing GFPmut3b (488 nm excitation, 520 nm emission) to the Cre-EBD of pCRE1 and pCRE4, respectively. Fluorescence was measured after the incubation of cells in "switch-on" media for 8 h. Yeast cultures were diluted to an OD600 of 0.1 with phosphate-buffered saline. Fluorescence of 50,000 individual cells were analyzed with a FACSCalibur flow cytometer (BD Biosciences). Data were processed using CellQuest Pro software (BD Bioscience). The "switch-on" media for pCRE5 and pCRE6 were SC-His medium containing 1 μM estradiol and SGal-His medium containing 1 μM estradiol, respectively. The "initial state" and the "switch-off" medium were SC-His medium. The GFP fluorescence of the cells was imaged from the cell lawn using a fluorescence microscope (Olympus CX41, Tokyo, Japan). The scale bars represent 10 μm.

**Deletion of target genes**. PCR protocol for PCR-mediated gene disruptions using primers in Supplementary Table 6. *His3* marker and *KanMX* can be amplified from the pRS413 and pFA6a-kanMX6 plasmids, respectively. This protocol usually results in amplification of only the specific product. Two hundred nanograms of gel-purified DNA fragment were directly transformed yeast.

**Screening of a random SCRaMbLE library**. First, the carotenoid pathway was integrated into the chromosome synV. The parent synV strains were transformed using the tight switch pCRE4 and plated onto SC-His agar. The plates were incubated at 30 °C for 72 h. Second, single colonies were inoculated into 5 ml SC-His overnight. The cultures were washed twice with ddH₂O and re-inoculated to obtain an OD600 of 1.0 in 2% galactose SGal-His medium containing 1 μM estradiol (Sigma-Aldrich). The cultures were incubated at 30 °C for 8 h to turn on Cre activity in the cells and begin the SCRaMbLE progress. Third, yeast cells were isolated from 1 ml of culture by spinning at 4000 × g in a centrifuge. The yeast pellet was washed twice with ddH₂O and resuspended in 1 ml of SC-His glucose medium. Fourth, the SCRaMbLE library was plated on SC-His glucose agar and the strains incubated at 30 °C until clear differences in carotenoid pigmentation were observed (72–120 h). Fifth, colonies were selected from the agar plates. Colonies colored darker than the wild-type parent strain were inoculated into a 48-well plate on SC-His glucose medium and dropped on SC-His glucose agar plates at 30 °C for 48 h. Sixth, strains colored darker than the wild-type parent strain in the 48-well plate were prepared for fermentation. Three independent colonies of each were inoculated in 5 ml of YPD medium for 24 h, then re-inoculated to obtain an OD600 of 0.1 in 40 ml YPD with 40 g l$^{-1}$ glucose medium in 250 ml flasks, which were then incubated for 60 h at 30 °C.

Carotenoids analysis. Caroteoids were extracted as described below. Cells collected from cultures were resuspended in 3 N HCl for boiling 2 min and cooled in an ice bath for 3 min. Then, cells debris were washed twice with water, resuspended in acetone containing 1% BHT (w v$^{-1}$), vortexed until colorless, followed by centrifugation. The acetone phase containing the extracted carotenoid was filtered for HPLC analysis. A HPLC system (Waters e2695) equipped with a BDS Hypersil C18 column (4.6 × 150 mm, 5 μm) and a UV/VIS detector (Waters 2489) was used to analyze the produced carotenoid. The signals of carotenoids and lycopene were detected at 450 and 470 nm, respectively. The mobile phase consisted of methanol-acetonitrile-dichloromethane (9:40:1 v v$^{-1}$) with a flow rate of 0.3 ml min$^{-1}$ at 30 °C.

**MuSIC experiments**. For iterative evolution cycles of diploid (synIII and synV), the yJBD000 was first generated by mating synV haploid with synIII haploid. The yJBD001 was selected from SCRaMbLE library of yJBD000; the yJBD038 was selected from SCRaMbLE library of yJBD001; the yJBD048 was selected from SCRaMbLE library of yJBD038; the yJBD057 was selected from SCRaMbLE library of yJBD048, and the yJBD069 was selected from SCRaMbLE library of yJBD057.

For incorporation of new synthetic chromosomes, the spore yJBS001 was generated from dissection of yJBD001. The yJBD200 was generated by mating yJBS001 (synIII and synV) with synX haploid. The yJBD201, yJBD202, yJBD203, yJBD204, and yJBD205 were all selected from SCRaMbLE library of yJBD200.

**Quantitative real-time PCR**. To assay copy numbers of duplications, yeast genomic DNA was used for qPCR analysis. The *ALG9* gene and the *YEL071W*, *CrtE*, *YEL022W*, *YER043C*, and *YER036C* genes were chosen as the reference gene and target genes, respectively. The copy numbers were quantified by comparing the Cq values of the target genes and the reference gene using the $2-\Delta\Delta Ct$ method[64]. Fast SYBR Green Master Mix (Applied Biosystems) was used for real-time PCR, and experiments were performed on the StepOnePlus Real-Time PCR System (Applied Biosystems). The oligonucleotide primers for qPCR are listed in Supplementary Table 6.

**Whole-genome sequencing and analysis**. All the deletions/duplications observed in this study were list in Supplementary Table 7. Deep sequencing and long-read sequencing were performed at Beijing Novogene Bioinformatics Technology Co., Ltd. Long-read single-molecule, real-time (SMRT) sequencing technology from Pacific Biosciences (PacBio) was used to sequence yJBH001 and the yJBS001. Genomic DNA was extracted. The collected DNA was detected by agarose gel electrophoresis and quantified by Qubit. Libraries were prepared according to the manufacturer's instructions for the PacBioz Template Prep Kit (Pacific Biosciences 10 kb template preparation protocol). The low-quality reads were filtered by SMRT 2.3.0[65,66] and the filtered reads were assembled to generate one contig without gaps. Deep sequencing of all libraries was performed on the Illumina HiSeq 2000 platform. The original figure data obtained by high-throughput sequencing were stored in FASTQ (fq) format, including the sequencing information and the corresponding sequencing quality information of the reads. The sequenced data were filtered and the adapter sequence and low-quality data were removed, resulting in the clean data used for subsequent analysis. The read comparison was the basis of the resequencing analysis. The variation information on the sample and the reference was obtained by aligning the sample reads with the designated reference. The reads were mapped to the reference sequence using BWA software and the coverage of the reference sequence with respect to the reads and explanation of the alignment results were obtained using the SAMTOOLS software. Structural variation (SV) refers to the insertion, deletion, inversion, and translocation of large segments at the genome level. The insertion (INS), deletion (DEL), inversion (INV), intra-chromosomal translocation (ITX), and inter-chromosomal translocation (CTX) between the reference and the sample were found by the BreakDancer software.

**PCRTags analysis**. Amplification of PCRTags was performed using 2 × EasyTaq PCR SuperMix (TRANSGEN BIOTECH), 400 nM each of forward and reverse primers (Supplementary Table 8), and genomic DNA in a 10 μl final reaction volume. PCR was performed as follows: 95 °C/1 min, 35 cycles of (95 °C/20 s, 64 °C/20 s/− 0.3 °C/cycle, 72 °C/30 s) and a final extension of 72 °C/5 min. Detection of PCRTags was performed by gel electrophoresis.

**Transcriptional analysis**. Yeast cells (yJBH000 and yJBH012) were grown in YPD medium to an OD600 of ~ 3.0 (exponential phase). A 20 ml aliquot of the yeast culture was centrifuged at 8000 × g for 2 min at room temperature. After removal of the supernatant, the cell pellets were frozen with liquid nitrogen, and then RNA was extracted using the QIAGEN RNeasy Mini kit. RNA sequencing was carried out by Beijing Novogene Bioinformatics Technology Co., Ltd (China). KOBAS software was used to test the statistical enrichment of differential expression genes in KEGG Pathways[67,68]. Triplicate samples were used for transcriptional analysis.

**Sporulation and tetrad dissection**. Fresh diploid colonies were inoculated in to 5 ml of YPD media and grown overnight at 30 °C. A fresh 5 ml of presporulation medium was then inoculated to an OD of 1 and grown for 4–5 h at 30 °C. The entire culture was then washed twice in 10 ml of water and inoculated into 2 ml of sporulation media[36]. Finally, cells were added to a final optical density of 1.0 OD and incubated at 25 °C for 3–10 days or until spores were visible. Sporulated cells were suspended in 50 μl of a 1 × stock solution of zymolyase and incubated at 30 °C for 5 min, then transferred to ice and diluted with 150 μl of cold $H_2O$. Tetrads were dissected using a SporePlay tetrad dissection microscope (Singer Instruments) and isolated spores were grown on YPD plates.

**Statistics**. No statistical methods were used to predetermine sample sizes. The carotenoids pathway SCRaMbLEd yeast colonies were picked by different colony colors, which were pre-established. Data collection and analysis were not randomized nor performed blind to the conditions of the experiments. No data points were excluded. The data are presented as mean ± SD and statistical differences were determined using unpaired *t*-test. Statistical significance was set as *$P < 0.05$ and **$P < 0.001$. The GraphPad Prism 6 was used for the statistical analyses.

**Data availability**. The data that support the findings of this study are available from the corresponding author on request. Transcriptomes data have been submitted to Gene Expression Omnibus under accession number GSE108580. Whole-genome sequencing is available at Genome Sequence Archive (GSA) in BIG Data Center (http://bigd.big.ac.cn/) under accession code CRA000723.

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

## Acknowledgements

This work was funded by the National Natural Science Foundation of China (21750001, 21621004, and 21676192), the Ministry of Science and Technology of China ("973" Program, 2014CB745100), and the International S&T Cooperation Program of China (2015DFA00960). We thank Hui-Min Liu and Zi-Xi Chen for help on mating type switching of synX yeast and flow cytometer experiments. We thank Wen-Hai Xiao and Ming-Zhu Ding for help on transcriptome analysis and helpful discussion.

## Author contributions

B.J., Y.W., and B.-Z.L. contributed equally to this work. B.J., Y.W., B.-Z.L., L.A.M., J.D.B., and Y.-J.Y. designed the experiments. B.J., Y.W., H.L., S.P., and J.W. performed the SCRaMbLE experiments. J.W. and H.-R.Z. performed sporulation experiments. B.J., Y.W., B.-Z.L., L.A.M., H.L., N.J., M.S., Z.-X.X., D.L., Y.-X.C., B.L., X.L., X.Z., and H.Q. analyzed the data and discussed the results. B.J. and B.-Z.L. wrote the manuscript. H.Q., L.A.M., M.S., J.D.B., and Y.-J.Y. edited the manuscript. Y.-J.Y. supervised all aspects of the study.
