## [Peer Review File · Nature Communications]

Reviewers' comments:

Reviewer #1 (Remarks to the Author):

This manuscript considers a new approach for SCRaMbLE to increase production phenotypes primarily. The key advance here is a controllable circuit to temper the expression and off-target effects of Cre. This paper needs to include many more details and controls to assess impact.

Many details of the experiments are missing including the full media type used, the number of colonies tested, etc. For example, the text states that carotenoid genes in CAN1 loci can allow for canavanine selection—was this actually done? If so (or even if not), would be interesting to see a control with just canavanine selection without Cre induction.

The frequencies shown and the overall experiments must be conducted with BIOLOGICAL and not technical replicates. There may be major starting point bias involved (esp. for data on figures like Figure 3).

Beyond genomes, it would be helpful to see transcriptomics to assess the impact of the modifications.

A major finding in this work was gene amplification (esp. with the carotenoid pathway). Given that there is a high homology in this work (with the lox sites among others), serial subculturing of the selected clones and outgrowth should be done to test stability of the resulting strains.

Reviewer #2 (Remarks to the Author):

Jia et al. 2017: Enabling precise control to SCRaMbLE synthetic haploid and diploid yeast

In this paper, Jia et al. discuss the development and evaluation of an alternative plasmid to perform SCRaMbLE (Synthetic Chromosome Rearrangement and Modification by LoxP-Mediated Evolution), a tool that allows rapid generation of diversity in synthetic chromosomes carrying LoxP sites. The paper can be subdivided in two parts. The first describes the performance of the new plasmid compared to the one that is currently mostly used. The second describes a proof-of-concept experiment, in which the authors test the applicability of the plasmid in haploid and diploid yeast to improve carotene production.

While the paper contains some interesting elements, there are quite a few serious problems with the experimental setup, data analysis and manuscript editing that need to be resolved. Also, we partly question the novelty, and feel that the added value of the presented data to the scientific community might be insufficient for publication in Nature Communications. The most pressing issues are listed below.

Major

- 1- In general, the manuscript format, grammar and language needs to be improved. There are numerous spelling and grammatical mistakes, not only in the text but also in the figures (eg 'glocuse', 'galatcose', ...). Moreover, it is overall very sloppy, often uses incorrect terminology and there are inconsistencies in strain and gene names.
- 2- The material and method section is incomplete. For example, there is no mention on how long-read sequencing is performed, how sporulation is done, how fluorescence experiments were performed and analyzed, how doubling time was calculated, what kind of 'Syn-yeast haploid' was used in MuSIC, etc etc.
- 3- The discussion is lacking depth and doesn't address the interesting or unexpected findings of

the study at all, and does not tie back the observations to previous scramble experiments.

4- The plasmid design described in this study is not new. In Cheng et al 2000 (NAR), the same construct was developed (called pSH62-EBD).

5- There seems to be a discrepancy between figure 1A and S2. As far as I can tell (but it is not very clear from the legend or Materials and Methods section), these are the exact same experimental conditions. However, in the figure in supplements, there is hardly any difference between viability of strains carrying pCRE1 and pCRE4, while this is the main point of Figure 1a.

6- The fluorescence experiment is not clear. The whole point of the SCW promoter is that it is only activated once in a cell's lifetime. This means that after 8 hours of growth in the 'on' phase, a lot of the cells in the population (all mothers) already had the Cre-spike, and are thus not supposed to have any expression anymore. The strain containing the gal-promoter on the other hand will be constitutively on. How is the fluorescence measured exactly (not mentioned in M&M), is it averaging out the value of all cells in the population? How big is the spread of fluorescence (please show histogram)? Based on the currently presented data, the authors can not claim that the new system "can be extremely induced to a higher expression level than pSCW11, resulting in an increased saturation of loxP sites in the synthetic chromosome by Cre-EBD". This last sentence also needs revision.

7- It is disappointing that no attempts to identify causative genes were made. Even for the scrambled haploid strains, where the authors narrowed the locus down to two genes, there was no attempt to unravel the genetic mechanism of improved carotene production. Also, in the experiment with the diploids and segregants, it remains unclear whether the observed improvement is solely due to the observed increase in copy number of the three carotene-genes that were integrated, or whether the other alterations also have an effect. The very least that should have been done in this experiment is increase the copy number of the carotene genes in the diploid (by simply introducing two copies), and investigate what the effect on carotene production is. Also, it would be worth checking whether the other improved selected diploids carry the same duplication.

8- It is hard to estimate whether the lack of leakiness of the GAL promoter is due to the absence of galactose or presence of glucose. For some application of scramble, researchers might not be able to use glucose in their medium, so it would be good to know whether the developed construct can also be applied in such conditions.

9- The quantification of colony colour (lower than control, normal and higher than control) is highly arbitrary, and the authors should come up with a more appropriate quantification. Next, a correlation between colour and carotene production should be established, to show that colour is indeed a good measure of production. Also, what is the reasoning of increasing T to allow better distinguishing between coloured colonies (please add reference)?

10- The low number of recombination events in one round of scrambling (eg. compared to Shen et al. 2016) is surprising. Please discuss. Is this due to usage of the GAL promoter? Control experiments with pCRE1 are missing.

11- It is very surprising that induction of sporulation induces more duplications and deletions (for example sporulation of yJBD001 -> yJBS001). It almost looks like scramble was induced during the sporulation process. This ties back to comment 8: the absence of glucose (at least I assume there is no glucose in the sporulation medium, it is not in the M&M) might be responsible for leakiness of the GAL promoter, resulting in another round of scrambling during sporulation. Also, this could explain why on average, the authors only observe a spore viability of ~25%, while it would be expected that spore viability is 50% (since there is only one region with essential genes deleted). Additional experiments need to be performed, and it should be addressed in discussion.

12- One interesting yet completely ignored observation is that the ratio lycopene/beta-carotene is highly diverse between scrambled strains. For example, in Figure 2F, this ratio is completely different in yJBN006 and yJBN008, even though the authors argue that deletion of the same gene (YEL014C or YEL013W) is responsible. Maybe this is an indication that the other region (YER042W) does have an effect on carotene production? Please discuss.

13- A non-scrambled control in the MuSIC experiment is missing.

14- In the MuSIC experiment, are there any alternations in the non-synthetic part of the chromosome (eg SNPs or duplications in the 9 genes in the MVA pathway)? The strategy involves

multiple rounds of selection, so it is not unthinkable. It should at least be checked and discussed.
15- A comprehensive table/figure with all deletions/duplications observed in all the improved scrambled strains would be useful.

Minor

- 1- The order of supplemental figures is sometimes confusing; please reorganize (or combine figures) to make it easier.
- 2- Some parts miss citations (eg. part on the genes involved in mevalonate pathway).
- 3- As it is a central part of the study, please elaborate a bit more on the leakiness of the pSCW-CRE-EBD system in the introduction.
- 4- Are any of the genes involved in the acetyl-coa to FPP pathway on synIII or synV? Please mention.
- 5- It is not very clear whether all the observed duplications and deletions are all flanked by loxP sites. In other words, are all observed changes due to scrambling? They probably are, but it should be mentioned. What about cross-overs and recombination during meiosis, do they have loxP breakpoints? That would be an interesting finding.
- 6- Is there any hypothesis why the diploid produces so much less carotene compared to the haploid (13-fold difference)?
- 7- Why did deep sequencing didn't pick up the CNVs in yJBD001, while it is so clear in yJBS001?
- 8- Fig3F: it is surprising that only ~30% of the diploid colonies showed no change in colony colour. This means that the vast majority of scrambled strains, which usually underwent only one or two recombination events, had a hit on genes that influence carotene production. This sounds unlikely, and should be discussed better (more elaborate than Lines 240-242).
- 9- Fig4E: add standard deviations.
- 10- Try to use consistent numbering of deleted/duplicated regions as much as possible. Eg. region 'c' in supplementary Fig22 is different than region 'c' in supplementary Figure 24 (where it is called region 'd'). This will make interpretation of the figures easier.
- 11- In the text, it is argued that yJBD057 has 4 additional duplications compared to yJBD048. This sounds counterintuitive, as it was previously shown that in one round of scrambling, on average one recombination event takes place. However, a look at supplementary figures 22 and 24 indicates that it is probably one large duplication event, spanning the region 0-120000. Please make more clear.
- 12- Compare the obtained carotene production with values obtained by other methods (metabolic engineering, mutagenesis, ...).

Reviewer #1:

Comment 1. Many details of the experiments are missing including the full media type used, the number of colonies tested, etc. For example, the text states that carotenoid genes in CAN1 loci can allow for canavanine selection—was this actually done? If so (or even if not), would be interesting to see a control with just canavanine selection without Cre induction.

Response: Thank you for the comment on the paper. We have revised details of the experiments and the material and method thoroughly, including full media types (line 524-534) and the number of colonies tested (Supplementary Table S5). The SC-Canavanine plate was used for selection of the carotenoids pathway integration. See Supplementary Fig. 5 and line 145-151.

Comment 2. The frequencies shown and the overall experiments must be conducted with BIOLOGICAL and not technical replicates. There may be major starting point bias involved (esp. for data on figures like Figure 3).

Response: Thank you for the comment. We have repeated the frequencies experiments with biological replicates. See Figure. 3f and Supplementary Table S5.

Comment 3. Beyond genomes, it would be helpful to see transcriptomics to assess the impact of the modifications.

Response: We sincerely appreciate the valuable comments. we have performed a transcriptomics to assess the impact of the modifications. See Supplementary Fig. 10, Supplementary Table S4 and line 201-216.

Comment 4. A major finding in this work was gene amplification (esp. with the carotenoid pathway). Given that there is a high homology in this work (with the lox sites among others), serial subculturing of the selected clones and outgrowth should be done to test stability of the resulting strains.

Response: Thank you for the valuable suggestion. We have assayed the stability of scrambled strains in serial sub culturing and revised the manuscript. See Supplementary Fig. 24 and line 400-404.

Reviewer #2

Major

Comment 1. In general, the manuscript format, grammar and language needs to be improved. There are numerous spelling and grammatical mistakes, not only in the text but also in the figures (eg 'glocuse', 'galatcose', ...). Moreover, it is overall very sloppy, often uses incorrect terminology and there are inconsistencies in strain and gene names.

Response: Thanks you. We have revised the manuscript and tried our best to improved the language thoroughly according to the comments. And here we did not list the changes but marked in red in revised paper.

Comment 2. The material and method section is incomplete. For example, there is no mention on how long-read sequencing is performed, how sporulation is done, how fluorescence experiments were performed and analyzed, how doubling time was calculated, what kind of ‘Syn-yeast haploid’ was used in MuSIC, etc etc.

Response: Thank you. We have revised the material and method thoroughly, including the long-read sequencing experiment (line 644-653), sporulation experiment (line 678-687), fluorescence experiments and analyzation (line 570-581), the doubling time calculation (line 558-568) and ‘Syn-yeast haploid’ was used in MuSIC (622-632).

Comment 3. The discussion is lacking depth and doesn’t address the interesting or unexpected findings of the study at all, and does not tie back the observations to previous scramble experiments.

Response: Thank you. We have revised the discussion thoroughly. We discussed some interesting findings (line 427-444) and compared the observations to previous scramble experiments (line 449-465).

Comment 4. The plasmid design described in this study is not new. In Cheng et al 2000 (NAR), the same construct was developed (called pSH62-EBD).

Response: We sincerely thank the reviewers for reminding us about the paper, we have checked the literatures carefully and added references in the revised manuscript.

Comment 5. There seems to be a discrepancy between figure 1A and S2. As far as I can tell (but it is not very clear from the legend or Materials and Methods section), these are the exact same experimental conditions. However, in the figure in supplements, there is hardly any difference between viability of strains carrying pCRE1 and pCRE4, while this is the main point of Figure 1a.

Response: We have measured the growth curves of synV strains containing the pCRE1, pCRE2, pCRE3, pCRE4 and the pRS413 in the SC-His medium. As shown in Supplementary Fig 2, the growth curves of strain containing the pCRE4 were nearly the same with control pRS413, the strain containing the pCRE1, pCRE2 and pCRE3 were observed growth defects compared to the control.

Comment 6. The fluorescence experiment is not clear. The whole point of the SCW promotor is that it is only activated once in a cell’s lifetime. This means that after 8 hours of growth in the ‘on’ phase, a lot of the cells in the population (all mothers) already had the Cre-spike, and are thus not supposed to have any expression anymore. The strain containing the gal-promotor on the other hand will be constitutively on. How is the fluorescence measured exactly (not mentioned in M&M), is it averaging out the value of all cells in the population? How big is the spread of fluorescence (please show histogram)? Based on the currently presented data, the authors can not claim that the new system “can be extremely induced to a higher expression level than pSCW11, resulting in an increased saturation of loxPsym sites in the synthetic chromosome by Cre-EBD”. This last sentence also needs revision.

Response: We sincerely appreciate the valuable comments. We used flow cytometer

to measure the fluorescence histogram of Cre-EBD-GFP expression derived by SCW11 promoter and GAL1 promoter (Fig 1d). We have revised the manuscript carefully. See line 124-136.

Comment 7. It is disappointing that no attempts to identify causative genes were made. Even for the scrambled haploid strains, where the authors narrowed the locus down to two genes, there was no attempt to unravel the genetic mechanism of improved carotene production. Also, in the experiment with the diploids and segregants, it remains unclear whether the observed improvement is solely due to the observed increase in copy number of the three carotene-genes that were integrated, or whether the other alterations also have an effect. The very least that should have been done in this experiment is increase the copy number of the carotene genes in the diploid (by simply introducing two copies), and investigate what the effect on carotene production is. Also, it would be worth checking whether the other improved selected diploids carry the same duplication.

Response: Thank you. We have revised the discussion thoroughly. We have proved that the deletion of YEL013W can increase the carotenoids production. Transcriptomics experiments were performed to assess the impact of the modifications. See 182-216 and 422-448. For duplication analysis, a diploid with two carotenoids pathway integration was generated and its carotenoid production was increased 4.2-fold compared with that of control but remained lower than that of yJBD001 (Fig. 3b). Four of the five scrambled diploids were observed to have carotenoid pathway duplication (Supplementary Fig. 13). This result indicated that some other SCRaMBLE events coordinate with the carotenoid pathway duplication to improve carotenoid production. See line 249-261.

Comment 8. It is hard to estimate whether the lack of leakiness of the GAL promotor is due to the absence of galactose or presence of glucose. For some application of scramble, researchers might not be able to use glucose in their medium, so it would be good to know whether the developed construct can also be applied in such conditions.

Response: According to reviewer's suggestion, we have used different carbon source to estimate whether the lack of leakiness of the GAL promotor. The yJBD000 strains containing the pCRE4 were cultured for 24 hours in SC-His medium containing 2% of glucose, sucrose, galactose, potassium acetate, ethanol and glycerol, respectively. As shown in Supplementary Fig 18 and line 345-350.

Comment 9. The quantification of colony colour (lower than control, normal and higher than control) is highly arbitrary, and the authors should come up with a more appropriate quantification. Next, a correlation between colour and carotene production should be established, to show that colour is indeed a good measure of production. Also, what is the reasoning of increasing T to allow better distinguishing between coloured colonies (please add reference)?

Response: According to reviewer's comment, we have assayed the relationships of colonies colors with the carotenoid production, five scrambled colonies exhibited

different colors were inoculated on SC-His glucose medium and then dropped on an SC-His glucose agar plate for comparison. The color of the strain was assigned the score of 1 and 5, respectively. Strains in the SCRaMbLE library were high-throughput screened by comparing colonies color to the 5 scores. See Supplementary Fig 7 and line 157-166. We have revised the manuscript and add references on the high temperature assist selection. See line 363-369.

Comment 10. The low number of recombination events in one round of scrambling (eg. compared to Shen et al. 2016) is surprising. Please discuss. Is this due to usage of the GAL promotor? Control experiments with pCRE1 are missing.

Response: The main different between this manuscript and Shen *et.al.* is the topological structure of chromosomes and the selection target. See line 456-465 for more discussion.

Comment 11. It is very surprising that induction of sporulation induces more duplications and deletions (for example sporulation of yJBD001 -> yJBS001). It almost looks like scramble was induced during the sporulation process. This ties back to comment 8: the absence of glucose (at least I assume there is no glucose in the sporulation medium, it is not in the M&M) might be responsible for leakiness of the GAL promotor, resulting in another round of scrambling during sporulation. Also, this could explain why on average, the authors only observe a spore viability of ~25%, while it would be expected that spore viability is 50% (since there is only one region with essential genes deleted). Additional experiments need to be performed, and it should be addressed in discussion.

Response: We sincerely appreciate the valuable comments. However, we did not think that SCRaMbLE was induced during the sporulation process. Because the average sequencing depth of the diploid strain (yJBD001) for the first time was about 200 X, which may not show structure variety clearly. We have sequenced the yJBD001 again and increased average sequencing depth to about 500 X. As shown in Supplementary Fig 12, the main duplications of the spore yJBS001 (YEL072W-YEL071W, YEL070W-YEL060W and YEL027W-YEL022W) were also observed in diploid strain yJBD001. The low spore viability is probably caused by some unconventional crossing-over. This phenomenon is beyond the scope of this report and we will explore it in the future.

Comment 12. One interesting yet completely ignored observation is that the ratio lycopene/beta-carotene is highly diverse between scrambled strains. For example, in Figure2F, this ratio is completely different in yJBN006 and yJBN008, even though the authors argue that deletion of the same gene (YEL014C or YEL013W) is responsible. Maybe this is an indication that the other region (YER042W) does have an effect on carotene production? Please discuss.

Response: We sincerely appreciate the valuable comments. We have revised the manuscript appropriately. Our results indicated that the deletion of YER042W increased the beta-carotene percentage of carotenoid production in combination with

the YEL013W deletion. See Figure 2f and line 436-444.

Comment 13. A non-scrambled control in the MuSIC experiment is missing.

Response: According to reviewer's suggestion, we have added the MuSIC experiments and revised the manuscript. See line 383-392 and line 622-632.

Comment 14. In the MuSIC experiment, are there any alternations in the non-synthetic part of the chromosome (eg SNPs or duplications in the 9 genes in the MVA pathway)? The strategy involves multiple rounds of selection, so it is not unthinkable. It should at least be checked and discussed.

Response: According to reviewer's comment, we have checked the non-synthetic part of the chromosome. As shown in Supplementary Fig 25 and 26, no SNPs or duplications were observed in the 9 genes of the MVA pathway. See line 449-456.

Comment 15. A comprehensive table/figure with all deletions/duplications observed in all the improved scrambled strains would be useful.

Response: We sincerely appreciate the valuable comments. We have revised the figures and manuscript to summarize all the deletions/duplication observed in all the improved scrambled strains. See Figure 2e, Figure 5d and Supplementary Table S7.

Minor,

Comment 1. The order of supplemental figures is sometimes confusing; please reorganize (or combine figures) to make it easier.

Response: We have checked the literatures carefully and reorganized figures to make it easier, thank you. See Supplementary Fig 8, 12, 22 and 23.

Comment 2. Some parts miss citations (eg. part on the genes involved in mevalonate pathway).

Response: Thank you for the advice. We have added references into the part on the genes involved in mevalonate pathway and many other parts in the revised manuscript. See Ref 22-26 in the text.

Comment 3. As it is a central part of the study, please elaborate a bit more on the leakiness of the pSCW-CRE-EBD system in the introduction.

Response: We have revised the introduction of manuscript according to reviewer's comment. See line 51-70.

Comment 4. Are any of the genes involved in the acetyl-coa to FPP pathway on synIII or synV? Please mention.

Response: We have summarized the chromosome locus of all the 9 genes involved in the acetyl-coa to FPP pathway in the Supplementary Table S5. None of the 9 genes is involved in the synIII or synV.

Comment 5. It is not very clear whether all the observed duplications and deletions are all flanked by loxP sites. In other words, are all observed changes due to scrambling? They probably are, but it should be mentioned. What about cross-overs and recombination during meiosis, do they have loxP breakpoints? That would be an interesting finding.

Response: We sincerely appreciate the valuable comments. First, we have checked all the boundaries of the duplications and deletions. As shown in Supplementary Fig 25 a and b, all the duplications and deletions were flanked by the loxpsym sites, suggesting that all observed changes due to scrambling. However, sequencing data showed that the crossing over was not flanked by loxpsym sites (Supplementary Fig 25 c). It is because the crossing over events is the exchange of genetic material between homologous chromosomes that results in recombinant chromosomes during meiosis, which was independent on scrambling and loxpsym sites.

Comment 6. Is there any hypothesis why the diploid produces so much less carotene compared to the haploid (13-fold difference)?

Response: It may be because heterozygous diploid was generated by mating a carotenoids produced synthetic haploid with a normal haploid, the compatibility between host cells and heterologous pathways decreased due to the gene dosage changes in the cellular genetic background. (Springer, M., Weissman, J. S., and Kirschner, M. W. (2010) A general lack of compensation for gene dosage in yeast. *Mol. Syst. Biol.* 6, 1–8.).

Comment 7. Why did deep sequencing didn't pick up the CNVs in yJBD001, while it is so clear in yJBS001?

Response: It is because that the average sequencing depth of the diploid strain (yJBD001) for the first time was about 200 X, which cannot show structure variety clearly. We have sequenced the yJBD001 again and increased average sequencing depth to about 500 X. As shown in Supplementary Fig 12, the main duplications of the spore yJBS001 (YEL072W-YEL071W, YEL070W-YEL060W and YEL027W-YEL022W) were also observed in diploid strain yJBD001.

Comment 8. Fig3F: it is surprising that only ~30% of the diploid colonies showed no change in colony colour. This means that the vast majority of scrambled strains, which usually underwent only one or two recombination events, had a hit on genes that influence carotene production. This sounds unlikely, and should be discussed better (more elaborate than Lines 240-242).

Response: We think there are two reasons. First, as the carotenoids pathway was on the downstream of MVA pathway, which is one of the core pathways and will be influenced significantly by the yeast growth condition. The random one or two recombination events causing growth defects or global transcriptional modification may easily influence carotene production; Second, the white color colonies were

made up of the most of the colour changed diploid colonies, which were caused by deletion of the synthetic chromosome segment containing the carotenoids pathway.

Comment 9. Fig4E: add standard deviations.

Response: We have revised the figures and manuscript according to reviewer's comment. See Figure 4d.

Comment 10. Try to use consistent numbering of deleted/duplicated regions as much as possible. Eg. region 'c' in supplementary Fig22 is different than region 'c' in supplementary Figure 24 (where it is called region 'd'). This will make interpretation of the figures easier.

Response: We have revised the figures and manuscript according to reviewer's suggestion. See Supplementary Fig 12, 15, 22 and 23.

Comment 11. In the text, it is argued that yJBD057 has 4 additional duplications compared to yJBD048. This sounds counterintuitive, as it was previously shown that in one round of scramble, on average one recombination event takes place. However, a look at supplementary figures 22 and 24 indicates that it is probably one large duplication event, spanning the region 0-120000. Please make more clear.

Response: We sincerely appreciate the valuable comments. We totally agree with the reviewer that yJBD057 had a very larger duplication (130 kb) range from YEL072W to YEL012W, compared to yJBD048. We have revised the manuscript. See line 393-400.

Comment 12. Compare the obtained carotene production with values obtained by other methods (metabolic engineering, mutagenesis, ...).

Response: We sincerely appreciate the valuable comments. We have discussed the carotene production with many different methods. See line 479 to 485.

Thank you for your consideration and we look forward to hearing from you.

Ying-Jin Yuan, PhD (corresponding author)
Professor of the biochemical engineering
Tianjin University, Tianjin 300072
P.R. China
Tel: 86-22-27401288
Fax:86-22-27403389
E-mail: yjyuan@tju.edu.cn

REVIEWERS' COMMENTS:

Reviewer #1 (Remarks to the Author):

The authors have made substantial efforts to improve their manuscript. It is a bit troubling to see that they originally were not presenting biological replicates (a key standard premise of the field).

While improvements have been made, I am not convinced about the generalizability of this approach for improving yeast. In particular, Figure S24 supports the concern I had in the last review--namely, the inclusion of all of these lox sites will ruin strain stability. In fact, stability was lost within only a few subcultures. This seems to be a very limited application of the approach that will not improve strains in the long term.

Reviewer #2 (Remarks to the Author):

I feel that this manuscript has matured significantly and can be published. That said, the language is perhaps still not perfect.

Reviewer #1:

Comment 1. The authors have made substantial efforts to improve their manuscript. It is a bit troubling to see that they originally were not presenting biological replicates (a key standard premise of the field).

While improvements have been made, I am not convinced about the generalizability of this approach for improving yeast. In particular, Figure S24 supports the concern I had in the last review--namely, the inclusion of all of these lox sites will ruin strain stability. In fact, stability was lost within only a few subcultures. This seems to be a very limited application of the approach that will not improve strains in the long term.

Response: Thank you for the comment on the paper. To test whether or not “the inclusion of all of these lox sites will ruin strain stability”, we have added subculture experiment and PCRTag analysis of the chromosome stability. As shown in Supplementary Fig. 24, we conclude that the synthetic chromosomes were stable after 60 generations, thus the inclusion of lox sites will not ruin strain stability¹⁻⁷. For the high production strain, only 0.68% colonies were in white color, in which deletions of YEL072W-YEL060W were observed. This may be caused by self-homologous recombination due to the toxicity of high production of carotenoids, which is consistent with previous literature⁸⁻⁹. To reduce the toxicity issue, inducible promoters can be used for uncoupling of growth and carotenoids production¹⁰⁻¹². See line 487-505 and Supplementary Fig. 24.

Reviewer #2:

I feel that this manuscript has matured significantly and can be published. That said, the language is perhaps still not perfect.

Response: Thank you for the comment on the paper. We have carefully revised the language and the formats of the manuscript.

REFERENCE

1. Xie, Z.-X. et al. ‘Perfect’ designer chromosome V and behavior of a ring derivative. *Science* **355**, (2017).
2. Wu, Y. et al. Bug mapping and fitness testing of chemically synthesized chromosome X. *Science* **355**, (2017).
3. Shen, Y. et al. Deep functional analysis of synII, a 770-kilobase synthetic yeast chromosome. *Science* **355**, (2017).
4. Mitchell, L. A. et al. Synthesis, debugging, and effects of synthetic chromosome consolidation: synVI and beyond. *Science* **355**, (2017).
5. Zhang, W. et al. Engineering the ribosomal DNA in a megabase synthetic chromosome. *Science* **355**, (2017).
6. Annaluru, N., Muller, H. & Mitchell, L. Total Synthesis of a Functional Designer Eukaryotic Chromosome. *Science* **344**, 55–58 (2014).

7. Dymond, J. S. et al. Synthetic chromosome arms function in yeast and generate phenotypic diversity by design. *Nature* **477**, 471–476 (2011).
8. Verwaal, R. et al. High-level production of beta-carotene in *Saccharomyces cerevisiae* by successive transformation with carotenogenic genes from *Xanthophyllomyces dendrorhous*. *Appl Environ Microbiol* **73**, 4342–4350 (2007).
9. Verwaal, R. et al. Heterologous carotenoid production in *Saccharomyces cerevisiae* induces the pleiotropic drug resistance stress response. *Yeast* **27**, 983–998 (2010).
10. Xie, W. et al. Construction of a controllable β -carotene biosynthetic pathway by decentralized assembly strategy in *Saccharomyces cerevisiae*. *Biotechnology and Bioengineering* **111**, 125–133 (2014).
11. Xie, W., Lv, X., Ye, L., Zhou, P. & Yu, H. Construction of lycopene-overproducing *Saccharomyces cerevisiae* by combining directed evolution and metabolic engineering. *Metabolic Engineering* **30**, 69–78 (2015).
12. Chen, Y. et al. Lycopene overproduction in *Saccharomyces cerevisiae* through combining pathway engineering with host engineering. *Microbial Cell Factories* **15**, 113 (2016).

Thank you for your consideration and we look forward to hearing from you.

Ying-Jin Yuan, PhD (corresponding author)
Professor of the biochemical engineering
Tianjin University, Tianjin 300072
P.R. China
Tel: 86-22-27401288
Fax: 86-22-27403389
E-mail: yjyuan@tju.edu.cn